# Electrical Performance of 28 nm-Node Varying Channel-Width nMOSFETs under DPN Process Treatments

**DOI:** 10.3390/mi13111861

**Published:** 2022-10-29

**Authors:** Shou-Yen Chao, Wen-How Lan, Shou-Kong Fan, Zi-Wen Zhon, Mu-Chun Wang

**Affiliations:** 1Department of Electronic Engineering, Minghsin University of Science and Technology, Hsinchu 30401, Taiwan; 2Department of Electrical Engineering, National University of Kaohsiung, Kaohsiung 81148, Taiwan; 3Department of Reliability Engineering, Chingis Technology Corporation, Hsinchu 30078, Taiwan

**Keywords:** nMOSFET, high-k, nitridation, subthreshold swing, threshold voltage, channel width

## Abstract

The decoupled-plasma nitridation treatment process is an effective recipe for repairing the trap issues when depositing high-k gate dielectric. Because of this effect, electrical performance is not only increased with the relative dielectric constant, but there is also a reduction in gate leakage. In the past, the effect of nitridation treatment on channel-length was revealed, but a channel-width effect with that treatment was not found. Sensing the different nano-node channel-width n-channel MOSFETs, the electrical characteristics of these test devices with nitridation treatments were studied and the relationship among them was analyzed. Based on measurement of the *V_T_*, *SS*, *G_m_*, *I_ON_*, and *I_OFF_* values of the tested devices, the electrical performance of them related to process treatment is improved, including the roll-off effect of channel-width devices. On the whole, the lower thermal budget in nitridation treatment shows better electrical performance for the tested channel-width devices.

## 1. Introduction

With regard to the complexity increment of nano-node process manufacturing, each process recipe in the production line will greatly impact the yield or the throughput of integrated-circuit (IC) products [1,2]. Even though the hot 3-nm IC mass-production technology at the present stage adopts a fin metal-oxide-semiconductor field-effect transistor (fin MOSFET or FinFET) [3,4,5,6,7], providing the better gate controllability, the high-k gate dielectric [8,9] is still a useful material with which to increase the drive current, *I_ON_*, related to the high-speed performance of ICs. Below 3-nm node process, the multi-nano-sheet field-effect transistors [10,11,12,13] with gate-all-around modality are more suitable candidates. Using the high-k dielectric is still a good choice for maintaining a higher drive current than that achieved with silicon dioxide or silicon nitride [14].

Because the high-k gate dielectric still supports an excellent k-value, more so than silicon dioxide, reducing the disadvantages of the high-k dielectric, such as the numerous traps in the atomic deposition of gate dielectric and the interface between channel surface and gate dielectric, is important. Using the thin interfacial layer is a feasible way of decreasing the interface state density and strengthening the bonding between the gate dielectric and the Si-based channel surface. Moreover, optimal nitridation treatment allows repair of the oxygen vacancy or bulk traps in the gate dielectric. Possible and cost-effective nitridation treatments include post-deposition annealing (PDA) and decoupled-plasma nitridation (DPN) processes [15,16]. According to the published literature [17], the PDA process in is more impressive in terms of cost, but the DPN process seems better in electrical performance due to the larger nitrogen free radicals fixing the traps more effectively. The major variables of DPN treatment processes in plasma systems include the radio-frequency power, nitrogen concentration, and treatment temperatures. Because the complementary MOSFET (CMOSFET), composed of an n-channel MOSFET (nMOSFET) and a p-channel MOSFET (pMOSFET), is the mainstay and foundation stone of logic, it has applications in data communication, data processing, and high-performance computing (HPC) IC products. The nano-node device model [18], a device with short-channel effect performance [19] and reliability [20,21], is more well suited to the development of electronic-design-automation (EDA) software [22]. Because the commercial EDA software is a good tool for IC designers, ultra-large-scale-integration ICs can be precisely designed and completed. However, few reports in the literature mention the nano-node channel-width effect after process variation. In this work, we aim to vary the nitrogen concentration and treatment temperatures impacting the electrical performance of channel-width devices after the deposition of high-k gate dielectric. These efforts will help to establish a set of precise device models with regard to channel-width after process variation.

The paper is organized as follows: In Section 2, an outline of semiconductor processes and a flow chart of electrical measurements are presented. In Section 3, the main sensing electrical results for each tested device and the analysis of nano-node channel-width performance are discussed. Moreover, the lower thermal budget of DPN treatment, on the whole, demonstrates better electrical performance among the three tested process groups. Finally, a summary of the precious findings and conclusions of this work is presented in Section 4.

## 2. Outline of Semiconductor Processes and Measurement Establishment

Although 28 nm-node processes have been gradually adopted at the mature process level, nitridation treatment in high-k gate dielectric can still be adopted for the novel 3 nm-node manufacturing process with FinFET format. The gate-last process [23,24] in the front-end level is more favored in integration consideration due to avoiding source/drain (S/D) diffusion after high-temperature annealing, impacting the metal gate instead of the poly-silicon electrode. Thus, substituting the traditional front-end process with the gate-last is necessary, and using low-resistance aluminum as the metal gate can improve the gate delay and power consumption. Incorporating the high-k HfO_2_, ZrO_2_, or sandwiched HfO_2_/ZrO_2_/HfO_2_ as the gate dielectric is a good way to increase the drive current and reduce the gate leakage, compared with silicon oxide at the equivalent oxide thickness [25]. In conventional planar MOSFET manufacturing, the active area (AA) must be defined first. The well and *V_T_* adjustment implants, forming the N- and P-wells and controlling the feasible *V_T_* values, are followed continuously. The sacrificial oxide is grown first and then removed. The true gate oxide is thermally grown. Furthermore, the poly-gate electrodes are produced using with chemical vapor deposition (CVD), dry etch technology and suitable photolithography. To reduce the hot-carrier effect, the S/D extension implant is used. The sidewall spacer shape is deposited and etched. After that, S/D implants and high-temperature annealing are used. The pre-metal dielectric, to provide device isolation, is deposited using low-temperature CVD technology. In addition, a contact mask is applied to form the gate contact (CT) and S/D CT. In addition, the first metal (M1) mask, made using with copper damascene [26,27] plus chemical-mechanical polishing, is used to gain the desired metal format. Finally, passivation and pad-window processes are completed to monitor the front-end device performance. For the gate-last processes, the poly-gate is treated as a dummy gate. The interfacial layer, SiO_x_, is deposited first with rapidly thermal oxidation process, before the 24 Å physical thickness of high-k dielectric deposition using an atomic-layer deposition process. This process is beneficial because it reduces the interface state density between the high-k gate dielectric and the channel surface and indirectly increases channel mobility. Generally, the hafnium dielectric is deposited early on in the flow, before a sacrificial poly-gate is created. After the high-temperature S/D and poly-silicide annealing cycles, the dummy gate is removed and Al-gate electrodes are deposited last. The remaining back-end processes with single damascene copper layer as the first metal layer were followed.

The overall complexity and process costs are slightly escalated, but the increase in electrical performance and decrease in power consumption is impressive. In this work, the tested wafers with gate-last processes were of an engineering type, the back-end metal was completed before the first metal (M1) plus passivation and pad window processes were begun, as shown in Figure 1. The abbreviations, *W*, *L*, *SDE,* and n^+^, in Figure 1 are channel-width, channel-length, source/drain extension implant [28], and heavily doped S/D implant, respectively. The dog-bone layout [29] has two advantages: eliminating the risk of gate-electrode peeling and avoiding the serious corner rounding in photolithography. After depositing the high-k gate dielectric, the nitridation treatment with the designed nitrogen concentration and annealing temperature was followed. In this work, three kinds of nitridation process splits are itemized as DPN-I, II, and III, respectively. The process information for the high-k gate dielectric with three DPN treatments is listed in Table 1. The nitrogen flow in terms of percentage (8–16%) in DPN process took place in an inert environment consisting of argon as a gas mixture.

With respect to the electrical measurement, the Keysight B1500A instrument was applied to assist the electrical parameter extraction. For the electrical characteristics of the tested devices, the threshold voltage (*V_T_*) with constant current metrology, drive current (*I_ON_*), OFF-state current (*I_OFF_*), transconductance (*G_m_*), and sub-threshold swing (*SS*) are more important. For the 28 nm-node logic processes, the supplied voltage *V_DD_* of the core device was 0.8 V. The measurement methods for extracting the electrical parameters are shown in Table 2.

The threshold voltage [30,31] without the body effect is a key to determining the switch capability of MOSFETs, which can be expressed as
(1)$$V_{T}=\Phi_{ms}-\frac{Q_{ox}}{C_{ox}}-\frac{Q_{d}}{C_{ox}}+2\phi_{F}$$
where Φ*_ms_* is the work function difference (Φ*_ms_* = Φ*_m_* − Φ*_s_* for a metal gate on Si substrate), *Q_ox_* is the total oxide charge, *C_ox_* is the inverse gate capacitance, *Q_d_* is the depletion charge (*Q_d_* = −[2 ε_s_ *q* N_a_ 2*ϕ_F_*]1/2), ε_s_ is the substrate dielectric constant, N_a_ is the channel surface doping concentration and *ϕ_F_* = (*E_i_* − *E_F_*)/*q*, where *q* is the unit charge, *E_i_* is the intrinsic-Fermi energy and *E_F_* is the Fermi energy.

The *V_T_* value of MOSFET can be extracted using the maximum *G_m_* method or the constant *I_DS_* method [32]; when considering the testing speed in the manufacturing line, the latter is preferred. The constant current method to calculate the *V_T_* value at the subthreshold characteristics can be represented as
(2)$$I_{DS}=\frac{W}{L}\cdot \mu_{n}\cdot (C_{d}+C_{ox})\cdot {\left(\frac{kT}{q}\right)}^{2}\cdot \left(1-e^{-\frac{qV_{D}}{kT}}\right)\cdot \left(e^{\frac{q\cdot (V_{G}-V_{T})}{C_{r}\cdot kT}}\right)$$
where *μ_n_* is the channel mobility for nMOSFET, *k* is the Boltzmann’s constant, *C_r_* = [1 + (*C_d_* + *C_it_*)/*C_ox_*], and *C_it_* is the interface-state capacitance.

As the *V_G_* = *V_T_* and the *V_D_* = 50 mV, the *I_DS_* (nA) is close to 100·*W*/*L* [30]. For the drive current, it can be treated as the saturation current of nMOSFET, *I_DS_sat_*.
(3)$$I_{DS\_sat}=I_{ON}=\frac{W}{2L}\cdot \mu_{n}\cdot C_{ox}\cdot {\left(V_{GS}-V_{T}\right)}^{2}\cdot (1+\lambda \cdot V_{DS})$$
where λ is the channel length modulation parameter.

The drive current at the linear region, *I_DS_lin_*, can be given as
(4)$$I_{DS\_lin}=\frac{W}{L}\cdot \mu_{n}\cdot C_{ox}\cdot \left(V_{GS}-V_{T}-V_{DS}/2\right)\cdot V_{DS}$$

The transconductance *G_m_* [33] is a derivative from *I_DS_lin_* by *V_GS_* as *V_DS_* fixed.
(5)$$G_{m}=\frac{\partial I_{DS}}{\partial V_{GS}}|V_{DS}{\;}_{fixed}=\frac{W}{L}\cdot \mu_{n}\cdot C_{ox}\cdot V_{DS}$$

The subthreshold swing *SS* [34] coming from Equation (2) is
(6)$$SS={\left(\frac{d\log(I_{DS})}{dV_{GS}}\right)}^{-1}=2.3\cdot \frac{kT}{q}\cdot \left(1+\frac{C_{d}+C_{it}}{C_{ox}}\right)$$

If the short-channel effect is considered, the drain-induced barrier lowering (*DIBL*) value is a good index by which to denote this phenomenon.
(7)$$DIBL=\frac{V_{T\_lin}-V_{T\_sat}}{V_{DD}-0.05}$$
where *V_T_lin_* is the *V_T_* value at the linear region and *V_T_sat_* is the *V_T_* value at the saturation region.

## 3. Results and Discussion

The measured channel-width of the devices, at a fixed channel-length *L* = 0.07 μm, were 1.5, 1, 0.3, and 0.1 μm under different nitridation treatments. The other device, *W*/*L* = 1.5/0.09 (μm/μm), was treated as a reference. The measured performance was classified into three sub-sections to reveal the channel-width effect related to nitridation treatment.

### 3.1. I_ON_ and I_OFF_ Parameters

The drive current, *I_ON_,* strongly influences transistor speed, especially in high-performance computing ICs. After the electrical measurement, the electrical characteristics for four tested nMOSFETs, under three types of DPN treatments are shown in Figure 2 at *V_GS_* = 0.5 V, which is greater than *V_T_* value, lessening the channel-length modulation effect. The comparison between *I_ON_* at *V_DS_* = 0.8 V and *V_GS_* = 0.5 V is shown in Table 3. With regard to *I_ON_* values, the DPN-I process seems to provide a better contribution, especially as the channel width is narrowed down, except in the wide channel-width device. The reason for the higher drive current in the wide-channel-width device, under the three nitridation treatments, could be that the deposition of high-k gate dielectric must remove the dummy gate first; in this example, the gate electrode exhibits a shallow trench. The dense concentration of nitrogen free radicals has a greater chance of fixing the traps of the gate dielectric, but the probability of forming silicon nitride or oxy-nitride on the channel surface is raised only a little. Therefore, the drive current in this tested device as a whole is increased. However, as the channel width is decreased, the uniformity of implantation and repair is also reduced. The drive current maintaining the integrity of the channel surface is also lowered. As the annealing temperature at DPN-I and -III is the same, the drive current with the lower nitrogen concentration is better than that with the heavier. DPN-II treatment has the highest thermal budget, easily generating the nitrogen compounds degrading the channel surface roughness. Hence, the performance of drive current in these three treatments is not the best.

The biggest influence on the *I_OFF_* values, came from front-end device leakage, including gate leakage, S/D junction leakage, and channel punch-through effect if the channel length is small enough. In Figure 3, the *I_OFF_* curves with three treatments are shown at *V_G_* = *V_B_* = *V_S_* = 0 V. In terms of the measured characteristics, the characteristics of the *W*/*L* = 1.5/0.07 (μm/μm) device with the three, which we analyzed the branches of *I_G_*, *I_S_*, and *I_B_* current flows in detail and the *I_G_* value contributing to the leakage weight, is indeed larger, compared with the *I_B_* value. Table 4 is an example of *I_OFF_* values as *V_DS_* = 0.8 V and *V_GS_* = 0 V under three DPN treatments. The *I_S_* ratio usually cannot afford to be ignored due to the channel punch-through effect. If the channel width is shortened, all the *I_S_* values are also reduced.

### 3.2. V_T_ and G_m_ Performance

The threshold voltage *V_T_* is a good index for the illustration of the controllability of the gate electrode in MOSFET. Based on the measured data, as shown in Figure 4, the *V_T_* values under *V_D_* biases with DPN-I treatment showed the better performance, indirectly illustrating the higher drive current with this treatment in Figure 2. As the channel width was shortened, most of the *V_T_* values of the tested devices went down, which means the smaller channel width, as channel length is fixed and is easily turned on. The nitrogen concentrations under DPN-I and II treatment were the same, as were *V_T_* distributions at the larger channel widths, but not at the shorter channel widths. At higher nitrogen concentration during treatment, the variation in *V_T_* distribution were apparent. The higher annealing temperature seemed to increase the *V_T_* values, especially for the wide channel-width device. A possible reason is that the larger gate area endures more channel interface degradation due to a thermal budget that is beneficial to the movement of nitrogen free radicals. These free radicals probably form nitrogen compounds, damaging the integrity of the channel interface. Additionally, the *V_T_* value was strongly related to the bulk traps and interface defects [17]. For the smaller device areas, the contribution of bulk traps to the *V_T_* value was distinctly reduced. Thus, the *W*/*L* = 0.1/0.07 (μm/μm) device always has a lower *V_T_* distribution. In Figure 4d, the roll-off effect of the channel-width devices at *V_D_* = 0.05 V and *L* = 0.07 μm can be observed and is consistent with the published literature [35,36] due to the edge-gate-oxide thinning at the corner of the AA zone. The results with DPN-I show a smoother distribution. The *V_T_* values with the three treatments and at *V_D_* = 0.05 V are shown at Table 5. It seems that the heavier N_2_ concentration contributes the higher *V_T_* than the lower.

The transconductance, *G_m_,* can be represented as the change in the drain current divided by the small change in the gate/source voltage with a constant drain/source voltage. In the literature [33], typical values of *G_m_* for a small-signal field-effect transistor with a submicron process area were less than 30 millisiemens (mS). However, this variable is also influenced by the ratio of *W*/*L*, channel mobility, and gate capacitance. Based on the measured results, as shown in Figure 5, the trends of *G_m_* vs. *V_G_* under three DPN treatments were predictable. If the maximum *G_m_max_* is extracted to do the comparison, all of the *G_m_max_* values are less than 1 mS. If the measured *G_m_max_* is normalized, based on a ratio of *W*/*L* = 1/0.07 (μm/μm), the contribution of the *W*/*L* ratio can roughly be ignored and the relationship between *G_m_max_* and channel mobility plus gate capacitance can be understood, as shown at Table 6. The best *G_m_max_* after normalization was located at *W*/*L* = 0.1/0.07 (μm/μm) with DPN-III. The worst belonged to the *W*/*L* = 1.5/0.07 (μm/μm) device with DPN-I. As the channel-width is reduced, the transconductance capability is increased in principle no matter what the treatment is. Nevertheless, the transconductance performance is not simply related to one or two variables. Because of entering nano-node manufacturing world, a slight variation of photolithography and etching technologies affects the accuracy of channel length and width. Therefore, the normalization applied to erase the effects of the variation in contribution from the preceding technologies is feasible but does not fully exclude a contribution to the *G_m_* effect.

### 3.3. SS and DIBL Considerations

A smaller subthreshold swing indicates better channel control promoting *I_ON_*/*I_OFF_* ratio, which usually means less leakage, and less power consumption. Furthermore, the subthreshold slope is a feature of a MOSFET’s current–voltage characteristics, as shown in Figure 6. In the subthreshold region, the drain current behavior controlled by the gate electrode is analogous to the exponentially decreasing current of a forward biased diode. Thus, a plot of drain current versus gate voltage at fixed drain, source, and bulk voltages will represent nearly a log linear action in this MOSFET operating regime. As described by Equation (6), the ideal minimum *SS* value is about 60 mV/decade. Thus, using a FinFET device in manufacturing is a feasible choice to achieve the smaller *SS* values, around 70 mV/decade, quoting from the Reference [9], with high-k gate dielectric. However, if the surface roughness was not well formed, the *SS* values were still somewhat enlarged with a SiO_2_/SiON gate dielectric, as described in the Reference [6]. The final consequences of *SS* values with different tested devices are exhibited in Table 7. The *SS* value is also a good index for explaining the degree of interfacial defects. A lower *SS* value means better interface integrity. Most of the *SS* values in the DPN-I treatment showed a larger *SS* value as the tested device is fixed and these values in DPN-I are greater than those in other two. In the overall results, the *SS* value of DPN-III at *W*/*L* = 1.5/0.07 (μm/μm) was the smallest. The hypothesis is that the higher nitrogen concentration provides more repair capability in bulk traps of high-k gate dielectric and relatively increases the gate capacitance [17]. This can be inferred from the *I_ON_* current, as shown in Table 3.

For the *DIBL* effect, the desired value is as small as possible. As claimed by the published literature [37], a *DIBL* value close to 30 mV/V is more desirable, but a value of less than 100 mV/V still acceptable in logic circuit design. In terms of the calculated data, the higher *DIBL* values belong to the larger channel width (*W* = 1.5 μm) with DPN-II and DPN–III. The possible reason for this is that the higher thermal budget, or higher nitrogen concentration, degrades the surface interface integrity and partially enlarges the *V_T_* value at lower *V_D_* bias, as shown at Table 8. Because the channel length *L* is ranged at the nano scale, the *DIBL* effect is, of course, a bit serious to measure.

In circuit design, pursuing HPC ICs is a target for increasing execution speed and reducing power consumption. However, if the designers need the smaller drive current to retard the speed of circuit module, using the layout technology with narrow channel-width devices is a helpful approach. Hence, revealing the electrical characteristics of nano-node channel-width devices is necessary, especially for high-k/metal gate dielectric with nitrogen treatments. In light of these electrical analyses, electrical performance is strongly related to the geometric size of the gate terminal, allowing doping uniformity and repair performance in high-k dielectric, and to the thermal budget and the nitrogen concentration in the treatment. There is a minor contribution from the uniform controllability of the photolithography and etching technology when forming the desired *W*/*L* sizes. As a result, the trend of *I_ON_* values in Table 3 does not fully follow the ratio of *W*/*L* due to the variation of the *V_T_* factor. The *V_T_* extraction with constant-current method shows the 100-nA coefficient, referred to as C. Hu’s recommendation [30]. In the view of commercial companies, this coefficient for nMOSFET or p-channel MOSFET is little tuned to fit the consideration of standby current in ICs. However, following the specially defined coefficient, the *V_T_* difference after data extraction between both transistors is less than 25 mV. For the extremely small *V_T_* value, this difference may be important, but for most devices, this difference can be ignored.

According to the electrical performance of the channel-width devices with DPN treatments, on the whole, DPN-I treatment, providing the smallest thermal budget, seems better than the others. 

## 4. Conclusions

Most of the electrical variables for channel-width nMOSFETs at a fixed channel length reveal the electrical performance under different DPN treatments. Besides the top-view contour of gate size possibly influencing the uniformity of nitrogen doping concentration, and indirectly affecting the *V_T_* and *SS* value, the lower thermal budget in nitridation treatment seems to better benefit the major electrical performance. The heavier nitrogen concentration probably causes the worse integrity of channel surface interface, and indirectly increases the *V_T_* values as well as reduces the drive current. In the end, the roll-off effect of channel-width devices is also evident, to a small extent, due to oxide thinning at the corner of the AA zone. In the future work, the reliability of channel-width effect devices under nitridation treatments will be an important area of investigation for solid-device applications.

## Figures and Tables

**Figure 1 micromachines-13-01861-f001:**
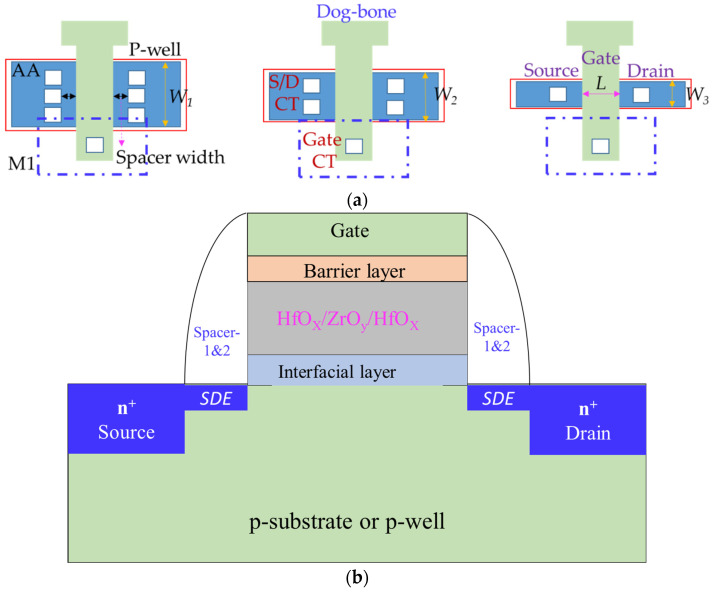
The schematic MOSFET with high-k/metal gate formation: (**a**) top-view layout as *W*_1_ > *W*_2_ > *W*_3_ and (**b**) cross-sectional profile.

**Figure 2 micromachines-13-01861-f002:**
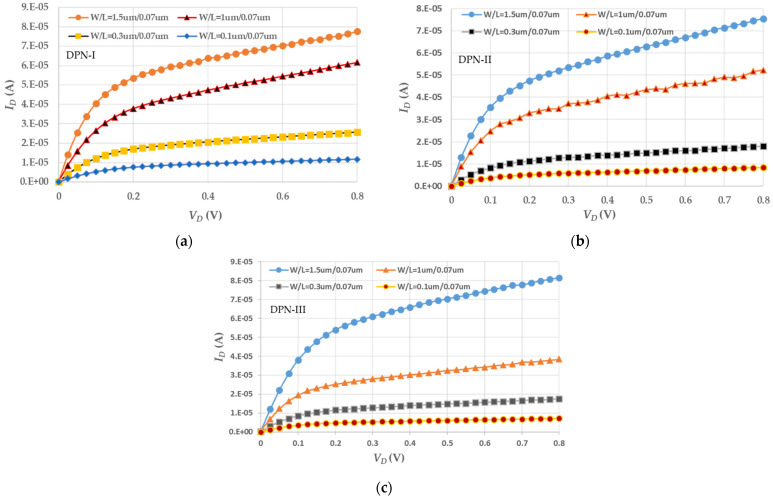
Electrical performance of *I_D_*–*V_D_* curves for four tested nMOSFETs (**a**) with DPN-I treatment, (**b**) with DPN-II treatment, and (**c**) with DPN-III treatment.

**Figure 3 micromachines-13-01861-f003:**
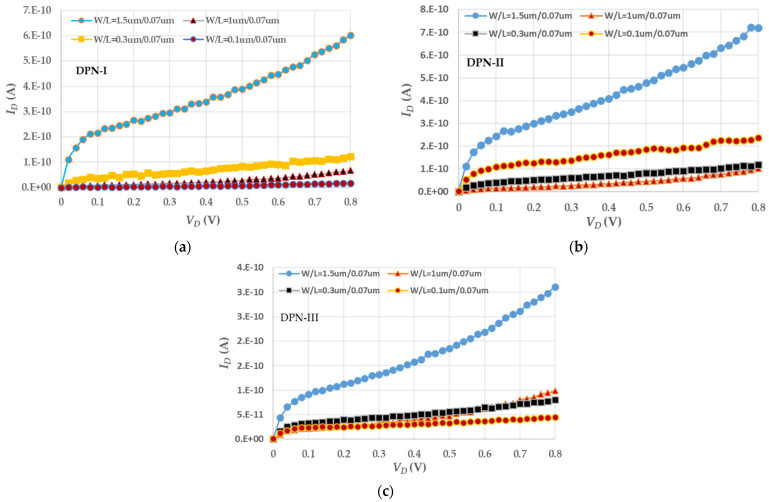
Electrical characteristics of *I_D_*–*V_D_* curves exposing OFF-state current (**a**) with DPN-I treatment, (**b**) with DPN-II treatment, and (**c**) with DPN-III treatment.

**Figure 4 micromachines-13-01861-f004:**
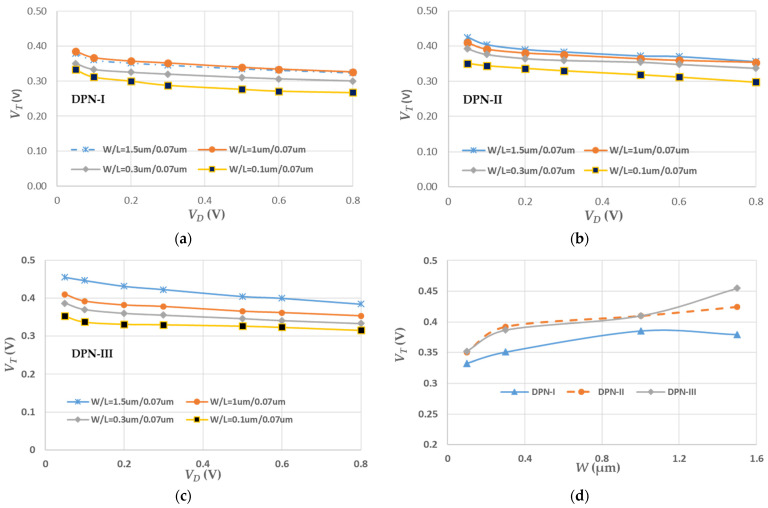
*V_T_* variables under different *V_D_* operations (**a**) with DPN-I treatment, (**b**) with DPN-II treatment, (**c**) with DPN-III treatment and (**d**) with channel-width effect at *V_D_* = 0.05 V and *L* = 0.07 μm.

**Figure 5 micromachines-13-01861-f005:**
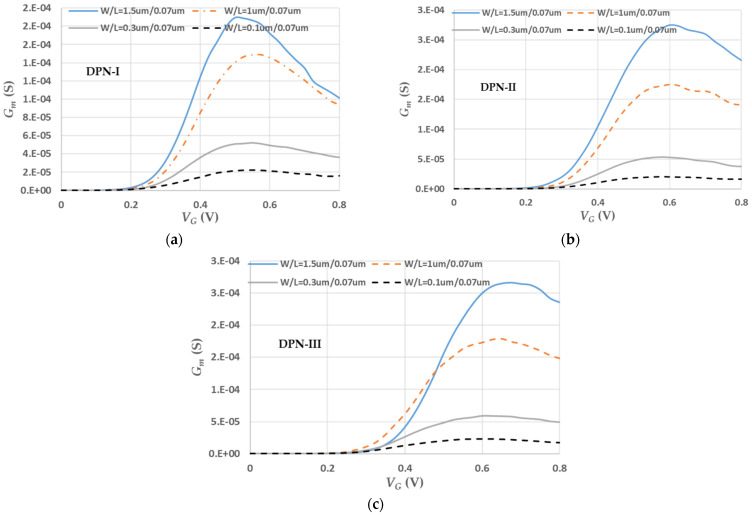
*G_m_* characteristics (**a**) with DPN-I treatment, (**b**) with DPN-II treatment, and (**c**) with DPN-III treatment.

**Figure 6 micromachines-13-01861-f006:**
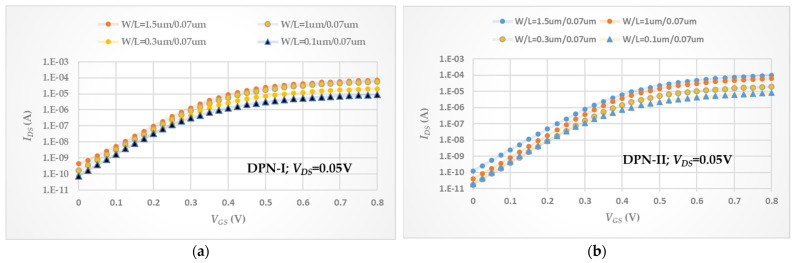
*I_DS_* vs. *V_GS_* performance with log-scale and *V_DS_
*= 0.05 V (**a**) under DPN-I treatment, (**b**) under DPN-II treatment, and (**c**) under DPN-III treatment.

**Table 1 micromachines-13-01861-t001:** Variables and parameters of gate dielectric deposition with three nitridation treatments.

No. Wafer	SiO_2_ (IL)	HfO_X_/ZrO_X_/HfO_X_ (Cycle)	Anneal	N_2_
DPN-I	9~12 Å	10/4/10	700 °C	8%
DPN-II	9~12 Å	10/4/10	900 °C	8%
DPN-III	9~12 Å	10/4/10	700 °C	16%

**Table 2 micromachines-13-01861-t002:** Valuable semiconductor parameters with sensing metrology.

Parameter Extraction	Sensing Metrology
*V_T_*	Sensing *I_DS_* − *V_GS_* as fixed *V_DS_* = 50 mV. Extracting the *V_T_* value as the expected *I_DS_* (nA) = 100 × *W*/*L*.
*I_ON_*	Measuring *I_DS_* − *V_DS_* as fixed *V_GS_* = *V_DD_* or (*V_GS_*−*V_T_*) = *V_DD_*. Choosing *I_DS_* as *V_DS_* = *V_DD_*.
*I_OFF_*	Sensing *I_DS_* − *V_DS_* as *V_G_* = *V_S_* = *V_B_* = 0 V. Recording *I_D_*, *I_G_*, *I_S_*, and *I_B_* as *V_DS_* = *V_DD_*.
*G_m_*	Deriving *I_DS_* − *V_GS_* as fixed *V_DS_* = 50 mV. Extracting the maximum *G_m_max_*.
*SS*	Deriving Log(*I_DS_*) − *V_GS_* as fixed *V_DS_* = 50 mV or *V_DD_*. Extracting the slope and taking the reciprocal.

**Table 3 micromachines-13-01861-t003:** *I_ON_* (μA) measured at *V_DS_* = 0.8 V and *V_GS_* = 0.5 V with three DPN treatments.

Tested Device *W*/*L* (μm/μm)	DPN-I	DPN-II	DPN-III
1.5/0.07	77.5	75.4	81.6
1/0.07	61.5	52.2	38.4
0.3/0.07	25.7	17.9	17.6
0.1/0.07	11.7	8.37	7.17

**Table 4 micromachines-13-01861-t004:** *I_OFF_* (pA) measured at *V_DS_* = 0.8 V and *V_GS_* = 0 V with three DPN treatments.

Tested Device *W*/*L* (μm/μm)	DPN-I	DPN-II	DPN-III
1.5/0.07	602	721	311
1/0.07	67.7	103	98.6
0.3/0.07	123	119	80.8
0.1/0.07	18	236	45.1

**Table 5 micromachines-13-01861-t005:** *V_T_* (V) measured at *V_DS_* = 0.05 V and *V_S_* = *V_B_* = 0 V with three DPN treatments.

Tested Device *W*/*L* (μm/μm)	DPN-I	DPN-II	DPN-III
1.5/0.07	0.380	0.425	0.455
1/0.07	0.386	0.410	0.410
0.3/0.07	0.351	0.392	0.387
0.1/0.07	0.333	0.350	0.352
1.5/0.09	0.368	0.362	0.396

**Table 6 micromachines-13-01861-t006:** *G_m_max_* (μS) measured at *V_DS_* = 0.05 V and *V_S_* = *V_B_* = 0 V with three DPN treatments with normalization.

Tested Device *W*/*L* (μm/μm)	DPN-I	DPN-II	DPN-III	DPN-I Normalization	DPN-II Normalization	DPN-III Normalization
1.5/0.07	190	274	267	127	183	178
1/0.07	149	175	179	149	175	179
0.3/0.07	52.3	53.1	59.2	174	177	197
0.1/0.07	22.0	20.8	23.2	220	208	232
1.5/0.09	204	235	123	175	202	105

**Table 7 micromachines-13-01861-t007:** *SS* (mV/decade) derived from *I_D_*–*V_G_* curves at *V_DS_* = 0.05 V and *V_S_* = *V_B_* = 0 V with three DPN treatments.

Tested Device *W*/*L* (μm/μm)	DPN-I	DPN-II	DPN-III
1.5/0.07	85.4	81.5	74.5
1/0.07	84.8	74.9	81.8
0.3/0.07	83.9	78.9	82
0.1/0.07	83	82.4	79.7
1.5/0.09	84.9	83.5	80.6

**Table 8 micromachines-13-01861-t008:** *DIBL* (mV/V) exposing the *V_T_* difference at *V_DS_* = 0.05 V and *V_DD_* with three DPN treatments.

Tested Device *W*/*L* (μm/μm)	DPN-I	DPN-II	DPN-III
1.5/0.07	74.1	92.1	94.4
1/0.07	79.7	75.5	76.1
0.3/0.07	67.5	73.2	71.2
0.1/0.07	87.3	70.0	49.6

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
