# Peer review of "Electrical Performance of 28 nm-Node Varying Channel-Width nMOSFETs under DPN Process Treatments"

_micromachines, 2022, doi:10.3390/mi13111861_

Round 1
Reviewer 1 Report
1. Needs to provide the results and discussion with variying different gate and drain voltage condition.
2. How does effect electron mobility for the variation of gate voltage?
3. Discuss about the changing of sub-threshold swing of the FinFET device.
4. How does contact resistance will effect where changing the channel length and width?
Author Response
Reviewer 1
- Needs to provide the results and discussion with varying different gate and drain voltage condition.
- How does effect electron mobility for the variation of gate voltage?
- Discuss about the changing of sub-threshold swing of the FinFET device.
- How does contact resistance will effect where changing the channel length and width?
Reply: Thank you for your positive guidelines.
Ans. 1: The IDS-VGS curves are added and indicated as Figure 6. The SS values in extraction are more richly demonstrated from the slope derivative. The measured curves indicated the different SS values, which meant the superiority of gate controllability and the integrity of surface interface. The correlations of SS values and these IDS-VDS curves are explained in sub-section 3.3. Thanks!
Ans. 2: As the gate voltage is increased, the moving carriers on the channel surface due to the surface roughness increased grow the scattering probability. Thus, the entire electron mobility is reduced. Basically, the DPN process is beneficial to high-k gate dielectric in bond repair, but the thermal budget is not controlled well. Some of nitrogen free radicals move to channel surface and chemically reacts with silicon to form silicon nitride. This material will increase the surface roughness. Thus, as the gate voltage is increased, more moving electrons will get the more scattering. Therefore, the channel mobility is reduced. Thanks for your concerns.
Ans. 3: In revised Reference 9, Aditya, M. et al have addressed the electrical characteristics of FinFETs with high-k gate dielectric. This article is a good reference exposing the SS values in FinFET due to the greater gate controllability. Following this article, it seems the SS values for FinFETs are lower than those for planar MOSFETs. We also add this explanation in Lines 270 to 274. In the past, this team also published over 50 peer-reviewed papers related to the FinFET topics. We also add two published articles as revised references 6 and 7 with SiO2/SiON gate dielectric.
Ans. 4: You are an expert in this area. Yes! The contact resistance, in the past, didn’t impact the source/drain current because it was smaller. However, as the channel length or channel width is narrowed down, this effect is probably concerned. As I was a senior device manager/ Technology Development Division/ UMC, I did the experiment and gained the contact resistance with Kelvin structure and four-probe technology. The probe contact resistance was less than 0.1 Ohm at that time. Thus, I believe that this side effect is not significant to impact the recently electrical measurement consequences. Thanks for your notice.
Ultimately, we really appreciate your kindly guidelines.
Reviewer 2 Report
Comments to the authors:
This manuscript makes a comparative analysis of decoupled plasma nitridation recipes to improve the performance of MOSFETs with high-k dielectric. Since most state-of-the-art transistors employ high-k materials as gate dielectrics, this type of analysis can be interesting to many readers. However, I would like to suggest the following improvements before this article can be published.
1) The authors should thoroughly revise this article to improve the structure of sentences and phrasing. In the current state of the article, the grammatical mistakes prevent the readers from fully understanding the content. The abstract and the introduction needs to undergo extensive revision for the message to be coherent. Also, the paper title has the term “DPN Nitridation,” which contains a redundancy, since “N” in DPN already stands for “nitridation”. Please take these issues under consideration.
2) In section 2, the authors provide a detailed description of both the gate-first and gate-last processes. Such a detailed description of the fabrication process is not relevant to the content of this article. The authors are requested to keep this description brief. For example, the authors can fully exclude the part describing the contact formation and metallization process.
3) In section 2, it is not mentioned clearly at what point during the fabrication flow the DPN process should take place. It should not be up to the readers to derive this information from the context. Much information is not provided about the DPN recipe either, which is the main topic of this paper. For example, the authors need to mention the gas flow sccm, plasma power and other tool information.
4) The authors compare three different nitridation recipes in terms of MOSFET characteristics. The comparative study shows the change in characteristics due to various nitridation recipes, but it does not provide any insights regarding the source of the change. The authors need to provide some qualitative (and quantitative, if possible) analysis on the effect of N2 concentration and anneal temperature on the bulk traps/ interface defects, in order to justify the validity of the characteristics change for each DPN recipe. For example, capacitance-voltage data from MOSCAPs could be used to provide insight into what happens at the interface.
Author Response
Reviewer 2
This manuscript makes a comparative analysis of decoupled plasma nitridation recipes to improve the performance of MOSFETs with high-k dielectric. Since most state-of-the-art transistors employ high-k materials as gate dielectrics, this type of analysis can be interesting to many readers. However, I would like to suggest the following improvements before this article can be published.
1) The authors should thoroughly revise this article to improve the structure of sentences and phrasing. In the current state of the article, the grammatical mistakes prevent the readers from fully understanding the content. The abstract and the introduction needs to undergo extensive revision for the message to be coherent. Also, the paper title has the term “DPN Nitridation,” which contains a redundancy, since “N” in DPN already stands for “nitridation”. Please take these issues under consideration.
2) In section 2, the authors provide a detailed description of both the gate-first and gate-last processes. Such a detailed description of the fabrication process is not relevant to the content of this article. The authors are requested to keep this description brief. For example, the authors can fully exclude the part describing the contact formation and metallization process.
3) In section 2, it is not mentioned clearly at what point during the fabrication flow the DPN process should take place. It should not be up to the readers to derive this information from the context. Much information is not provided about the DPN recipe either, which is the main topic of this paper. For example, the authors need to mention the gas flow sccm, plasma power and other tool information.
4) The authors compare three different nitridation recipes in terms of MOSFET characteristics. The comparative study shows the change in characteristics due to various nitridation recipes, but it does not provide any insights regarding the source of the change. The authors need to provide some qualitative (and quantitative, if possible) analysis on the effect of N2 concentration and anneal temperature on the bulk traps/ interface defects, in order to justify the validity of the characteristics change for each DPN recipe. For example, capacitance-voltage data from MOSCAPs could be used to provide insight into what happens at the interface.
Reply: Thank you for your constructive feedback and encouragement.
Ans. 1: Thanks for your comments to abound in this article. We accept your advice and change the title a little. The relationship between abstract and introduction is improved more, too. Besides revising this content again, the revised content in English is entrusted by MDPI Editage author service to help us improve the English quality. The payment in English editing is also attached in the last page. Thanks!
Ans. 2: Thanks for your advice. However, as we checked the description in section 2 again, we didn’t mention too much comparison between the gate-first and gate-last processes. If we exclude the contact formation and the metallization processes until the first metal (M1), mentioned in Lines 90-93, the gate-last processes until M1 in Line 101 must be addressed again. Thus, if we did that, it seems the readers can’t catch the entire gate-last process flow and confuse the Figure 1(a) in this work. Hence, after the team’s discussion, we still suggest keeping this part better. Thanks for your consideration.
Ans. 3: You are indeed an expert to emphasize the key parameters of DPN processes. For these process parameters of gas flow sccm, plasma power and other tool information in DPN processes, the data information belongs to UMC. We tried to ask them for sharing those, but UMC said they were commercial confidential and declined our requests to share them. Thus, we really don’t know what to answer your concerns. We apologize that. For the DPN process steps, we add them in Lines 110-112. Thanks for your notice.
Ans. 4: Thanks for your good notice and great advice. Your comments are indeed necessary to richly demonstrate the DPN effect in gate dielectric. We can add some comments of the insights regarding the sources of the change. Based on your concerns, in revised Reference 17, this team had exposed the gate leakage, bulk traps and interface defects with the MOSCAPs, but didn’t analyze the channel-width effect with DPN treatment. Thus, we chose this topic in deep investigation and compensated the weakness in the establishment of device models. We have added the explanation in Lines 270-282 to qualitatively explain the interface defects impacting the SS values and the bulk traps influencing the VT value in Lines 221-224.
In the end, many thanks for your positive advice and encouragement.
Round 2
Reviewer 2 Report
Comments to the Author
Thanks to the authors for addressing most of the comments/ concerns from this reviewer carefully. The reviewer would like to recommend a few important changes before publishing:
- The paper title and abstract should reflect that the DPN process is actually a post-deposition nitridation of the high-k gate dielectric. It is difficult for a casual reader to grasp this fact by glancing through the article. DPN can be also used for interfacial layer (IL) treatment, pre-deposition surface treatment etc. So, the functionality of this DPN process should be stated explicitly in the abstract and the paper title.
- The term ’long/short channel-width” can lead to a lot of confusion, since “long/short channel” predominantly refers to the length of a channel. This work mainly varies the channel width, but not the length. Hence authors are asked to substitute ’long/short channel-width” in the paper title with “varying channel width.” In the text, the authors can distinguish between devices with different widths as “wide-channel” or “narrow-channel” devices.
- Gate-last process is the typical one for state-of-the-art high-k gate stack at advanced CMOS nodes. Was gate-last process used to fabricate the devices used in this study? If that is the case, the entire discussion on the gate-first process is unnecessary and will confuse the readers. The benefits of the gate-last process over the gate-first process has been well-documented in the literature and does not need further elaboration. The reviewer suggests not mentioning the gate-first process at all and describing the relevant details of gate-last process only, in order to make this article straightforward and concise.
- How is the SiO2 IL grown? Is this a wet chemical oxidation process or a rapid thermal oxidation process? This detail is important since it contributes to the quality of the gate oxide and interface. Please add this information.
- Table I shows the nitrogen flow in terms of percentage (8-16%). Readers will wonder about gases consisting of the rest of the mixture. Is the DPN taking place in a inert environment consisting of argon? Please elaborate.
- Regarding the results, the reviewer would like to reiterate one important point they had raised in the previous round of reviews. The reviewer understands that parameters of gas flow sccm, plasma power in the DPN process are confidential information. Hence this paper cannot help the readers to reproduce the DPN conditions used here. However, this paper can help the readers understand the effects of varying different nitridation parameter, i.e. nitrogen concentration and anneal temperature. For example, please provide some qualitative insights into why the threshold voltage increased when nitridation temperature was increased. How does the increase of N2 concentration from 8% to 16% qualitatively change the interface and device performance? In summary, there should be some qualitative explanation of the differences among DPN-I, DPN-II and DPN-III, as seen in Table 3-6.
Author Response
Thanks to the authors for addressing most of the comments/ concerns from this reviewer carefully. The reviewer would like to recommend a few important changes before publishing:
- The paper title and abstract should reflect that the DPN process is actually a post-deposition nitridation of the high-k gate dielectric. It is difficult for a casual reader to grasp this fact by glancing through the article. DPN can be also used for interfacial layer (IL) treatment, pre-deposition surface treatment etc. So, the functionality of this DPN process should be stated explicitly in the abstract and the paper title.
- The term ’long/short channel-width” can lead to a lot of confusion, since “long/short channel” predominantly refers to the length of a channel. This work mainly varies the channel width, but not the length. Hence authors are asked to substitute ’long/short channel-width” in the paper title with “varying channel width.” In the text, the authors can distinguish between devices with different widths as “wide-channel” or “narrow-channel” devices.
- Gate-last process is the typical one for state-of-the-art high-k gate stack at advanced CMOS nodes. Was gate-last process used to fabricate the devices used in this study? If that is the case, the entire discussion on the gate-first process is unnecessary and will confuse the readers. The benefits of the gate-last process over the gate-first process has been well-documented in the literature and does not need further elaboration. The reviewer suggests not mentioning the gate-first process at all and describing the relevant details of gate-last process only, in order to make this article straightforward and concise.
- How is the SiO2 IL grown? Is this a wet chemical oxidation process or a rapid thermal oxidation process? This detail is important since it contributes to the quality of the gate oxide and interface. Please add this information.
- Table I shows the nitrogen flow in terms of percentage (8-16%). Readers will wonder about gases consisting of the rest of the mixture. Is the DPN taking place in an inert environment consisting of argon? Please elaborate.
- Regarding the results, the reviewer would like to reiterate one important point they had raised in the previous round of reviews. The reviewer understands that parameters of gas flow sccm, plasma power in the DPN process are confidential information. Hence this paper cannot help the readers to reproduce the DPN conditions used here. However, this paper can help the readers understand the effects of varying different nitridation parameter, i.e. nitrogen concentration and anneal temperature. For example, please provide some qualitative insights into why the threshold voltage increased when nitridation temperature was increased. How does the increase of N2 concentration from 8% to 16% qualitatively change the interface and device performance? In summary, there should be some qualitative explanation of the differences among DPN-I, DPN-II and DPN-III, as seen in Table 3-6.
Reply: Thank you for your positive guidelines and great comments.
Ans. 1: If you believe that the abstract may confuse the readers in nitridation treatment, we add “decoupled-plasma” in the abstract to eliminate this confusion. DPN is used for high-k gate dielectric. If the DPN is treated as IL treatment, the free radical of nitrogen is easily formed as silicon nitride and degrades the channel mobility due to the bad surface roughness. Thus, the drive current was possibly decreased, as shown in Table 3. We have exposed this weakness in Lines 182-185. We agree that if the suitable controllability of DPN process may be used for your mentions in these functionalities, we hope we will have a great chance to get the cooperation with UMC to discuss these interesting research topics. But, in this work, we aim at the DPN process in high-k gate dielectric. Thanks for your great advice.
Ans. 2: Your comment in title is reasonable. We follow your suggestion and modify the title. The “long channel-width” in text is changed as “wide channel-width”. For the “short channel-width”, we didn’t mention it. Thus, we didn’t change it as “narrow-channel”.
Ans. 3: Yes! These wafers were formed by the gate-last process. Following your comments, we remove the entire discussion on the gate-first process to avoid the confusion.
Ans. 4: SiO2 IL is grown with a rapid thermal oxidation process. We follow your advice and add it in Line 95.
Ans. 5: Yes! We follow your illustration and add it in Lines 116-117. Basically, the wafer provider forbade us to show much process information. However, you have guessed the process operation. We add it in the content.
Ans. 6: Thanks for your consideration for some commercial confidential at the first-round review. Basically, we have discussed the relationship between the treatment processes and the electrical performance, especially the ION and VT. This VT value is strongly related to interface-state density and oxide-trap density. According to the SS values, the interface state density can be qualitatively compared with the same geometric device size. Thus, it is possible to impact the VT and indirectly influence the ON current. We add the N2 percentage probably impacting the drive current in Lines 185-186 and add the contribution of N2 percentage in conclusion part in Lines 331-333.
Finally, we sincerely appreciate you for your great comments to make this article straightforward and concise.